# Subjective perception of visual field defects using random noise-moving images in patients with glaucoma: A comparison of computer graphics and analog noises

Arata Inoue [1]*, Eiko Koike[2], Naoyuki Maeda [3], Chota Matsumoto[4]

1 Inoue Eye Clinic, Sakai, Osaka, Japan, 2 Koike Eye Clinic, Sakai, Osaka, Japan, 3 Department of Ophthalmology, Osaka University Graduate School of Medicine, Suita, Osaka, Japan, 4 Department of Ophthalmology, Faculty of Medicine, Kindai University, Osaka-Sayama, Osaka, Japan

* iec_info@lion.ocn.ne.jp

## Abstract

### Purpose

Random noise-moving images (noises) can make glaucoma patients with no subjective symptoms aware of visual field abnormalities. To explore this concept, we developed a noise using computer graphics (CG) and investigated the difference in the subjective perception of visual field abnormalities between CG and conventional analog noises.

### Methods

We enrolled individuals with glaucoma (205 eyes), preperimetric glaucoma (PPG; 19 eyes), and normal eyes (35 eyes). For a CG noise, a series of still images was made by randomly selecting five monochromatic tones on 2-mm square dots, and these images were drawn at 60 frames per second (fps) to create a noise-moving image. The participants were asked to describe their perceived shadows on a paper. The results were categorized as follows based on the pattern deviation probability map of the Humphrey field analyzer (HFA): "agreement," "partial agreement," "disagreement," and "no response." The glaucoma stage was classified into four stages, from M1 to M4, based on the HFA's mean deviation.

### Result

The detection rates (agreement and partial agreement) were 80.5% and 65.4% for the CG and analog noises, respectively, with CG noise showing a significantly higher detection rate in all glaucoma eyes ($P < 0.001$). The detection rates tended to increase as the glaucoma stage progressed, and in Stage M3, these were 93.9% and 78.8% for the CG and analog noises, respectively. The PPG eyes did not exhibit subjective abnormalities for both noises. The specificity values were 97.1% and 100% for the CG and analog noises, respectively.

**Data Availability Statement:** All relevant data are within the paper and its Supporting Information files.

**Funding:** The author(s) received no specific funding for this work.

**Competing interests:** Arata Inoue has a patent for a method and device for diagnosing visual field abnormalities (patent No. JP7339900), as mentioned in this article. There are no other relevant declarations relating to employment, consultancy, further patents, products in development or marketed products. This does not alter our adherence to PLOS ONE policies on sharing data and materials. Eiko Koike: none. Naoyuki Maeda: Paid consultancy: BVI Medical, CooperVision, Tomey Corporation. Research Grant: Senjyu Pharmaceutical Co., Ltd., Topcon Corporation. Honoraria for speaking: Alcon, Inc., Bausch + Lomb, HOYA Corporation, Johnson & Johnson, Menicon Co., Ltd., Nikon Corporation, Novartis International AG, Otsuka Pharmaceutical Co., Ltd., Santen Pharmaceutical Co., Ltd., Topcon Corporation. Chota Matsumoto: Paid consultancy: CREWT Medical Systems, Inc., Topcon Corporation. Honoraria for speaking: CREWT Medical Systems, Inc., Topcon Corporation. There are no patents, products in development or marketed products to declare. These do not alter our adherence to PLOS ONE policies on sharing data and materials.

## Conclusion

The CG noise is more effective than the analog noise in evaluating the subjective perception of visual field abnormalities in patients with glaucoma.

## Introduction

Patients with glaucoma are often unaware of their visual field defects until the late stages [1], and their visual field impairment may be markedly advanced at the time of diagnosis [2]. Additionally, many patients discontinue topical treatments due to a lack of subjective symptoms. Therefore, making patients aware of visual field abnormalities may motivate them to visit medical facilities and thus assist in the early diagnosis of glaucoma. Furthermore, it may help patients understand their medical conditions, improve their adherence to topical therapy, and facilitate the acquisition of informed consent for surgery.

As print media are used to recognize visual field abnormality, the Amsler grid test [3–5], Suzuki's eye-check chart [6], and Clock Chart [7, 8] have been utilized in previous reports. For electronic media, it is known that when watching random noise-moving images (abbreviated as "noise") on a television (TV) or computer display, patients who are unaware of visual field defects can recognize their visual field abnormalities [9]. Patients can recognize areas that appear "nonblinking, cloudy, blackish, gray, or whitish," which are consistent with their visual field defects. The examination using this phenomenon has been called "noise-field/snow field campimetry" or "noise-field test." This test only requires patients to watch a noise. The abnormalities in the entire visual field can be confirmed in a few seconds, and a negative scotoma can be converted to a positive scotoma.

In 1987, Aulhorn reported that visual field defects could be recognized by a snow noise, that is, an analog noise, which occurs on an unbroadcast analog TV [9]. In 1989, a computer-controlled digital noise, called noise-field campimetry, was developed with the Tübingen electronic campimeter (TEC) [10], which has been extensively reported by Schiefer as very useful [9]. However, the production of TEC has already been discontinued.

For analog noise examination of home TVs, Shirato et al. reported its usefulness in 1991 [11], and Adachi et al. indicated that its detection rate was higher than that of TEC in 1994 [12]. A large-scale screening test using home TVs was also conducted, which was reported to be effective [13]. Currently, analog TV broadcasting has not been used in various countries, and the opportunity to use analog noise is almost gone.

Therefore, we created digital noises using computer graphics (CG) and compared their properties with those of conventional analog noises to determine which is more effective for the subjective detection of visual field defects.

## Methods

### Noise-field test

In the present study, the snow noise generated by unbroadcast analog reception TVs was used as an analog noise. As a digital noise, we developed a computer-controlled artificial noise made with CG (CG noise). The noise-field test with the analog and CG noises was conducted.

### Selection of analog noise

Before the study, we visually inspected many noises on 17 TVs and video recorders with built-in analog broadcast tuners and identified the most homogeneous and stable noise. As a result,

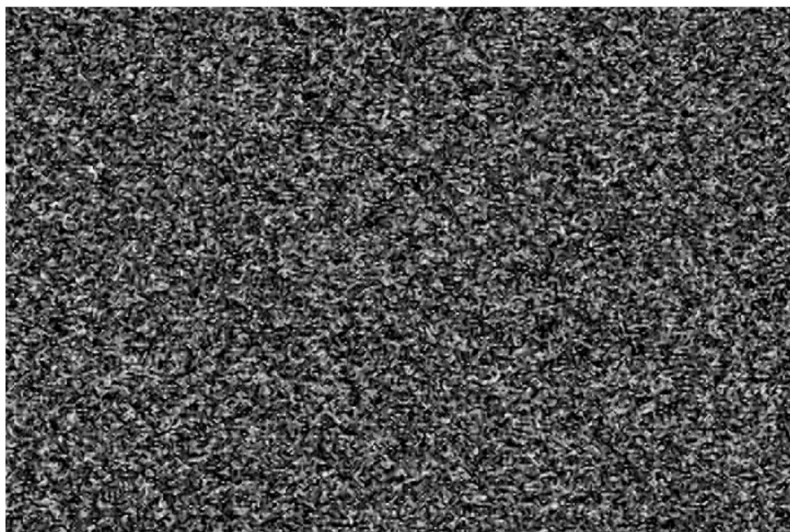 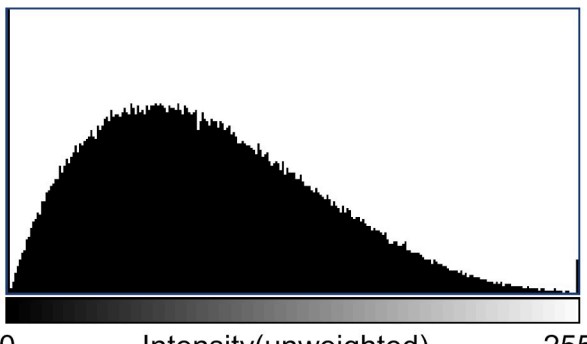

0    Intensity(unweighted)    255

N: 345600          Min: 0
Mean: 79.809       Max: 255
StdDev: 54.595

**Fig 1. Still image of analog noise.** (Left side) Still image of analog noise: the analog noise was recorded on a hard disk recorder and copied to a Blu-ray Disc®, and a still image (1 frame) was extracted on a personal computer using a video-editing software program from the Blu-ray Disc. The noise had a resolution of 720 × 480 (= 345,600) pixels, but it was displayed on full screen on the monitor. (Right side) A histogram of luminance distribution: all pixels' luminance value (gray value; 0–255) was measured using image analysis software. The histogram shows the luminance value on the X-axis and the number of pixels for each on the Y-axis, indicating that the luminance distribution shows a normal distribution-like bell-shape. N (total pixel count): 345,600, Mean (luminance value): 79.809, StdDev (standard deviation): 54.595.

a liquid-crystal display (LCD) TV (LC-20D10®, Sharp Corp., Sakai, Japan) without an antenna connection was selected (Fig 1).

## Production of CG noise

We installed software programs (detailed in Table 1) on a personal computer (PC: CF-SV7JDUQR®, Intel Core i7-8550U processor, Intel UHD Graphics 620, Panasonic Corp., Osaka, Japan; Operating system: Microsoft Windows 10®) to create the CG noise using the Visual C++ programming language (Microsoft Corp., Redmond, WA).

First, 2-mm square dots (exact size 7 pixels, 1.89 mm) with five different monochromatic tones were made, randomly selected using a random number generator (RNG), and placed sequentially from the edge of the screen to produce a single still image. Next, one still image was displayed, while the next one was generated simultaneously. When displayed sequentially at 16-millisecond intervals, a noise-moving image at 60 frames per second (fps) was obtained (Fig 2, S1 Video). The dot size, monochromatic tone, and frequency of the CG noise were set with reference to the TEC setting.

The density ratios of the five tones were specified as 1.0, 0.75, 0.5, 0.25, and 0.0 on the computer program (Fig 3). Using a luminance meter (SM208, Sanpo Instrument Co., Ltd,

**Table 1. Software environment for CG noise creation.**

| |
|---|
| OS: Microsoft Windows10®, OpenGL® (Graphics Library). |
| Microsoft Visual Studio Community® 2017, Visual C++ programming language |
| freeglut |

CG: computer graphics, OS: operating system
The freeglut is a free-software/open-source alternative to the OpenGL Utility Toolkit.
http://freeglut.sourceforge.net/

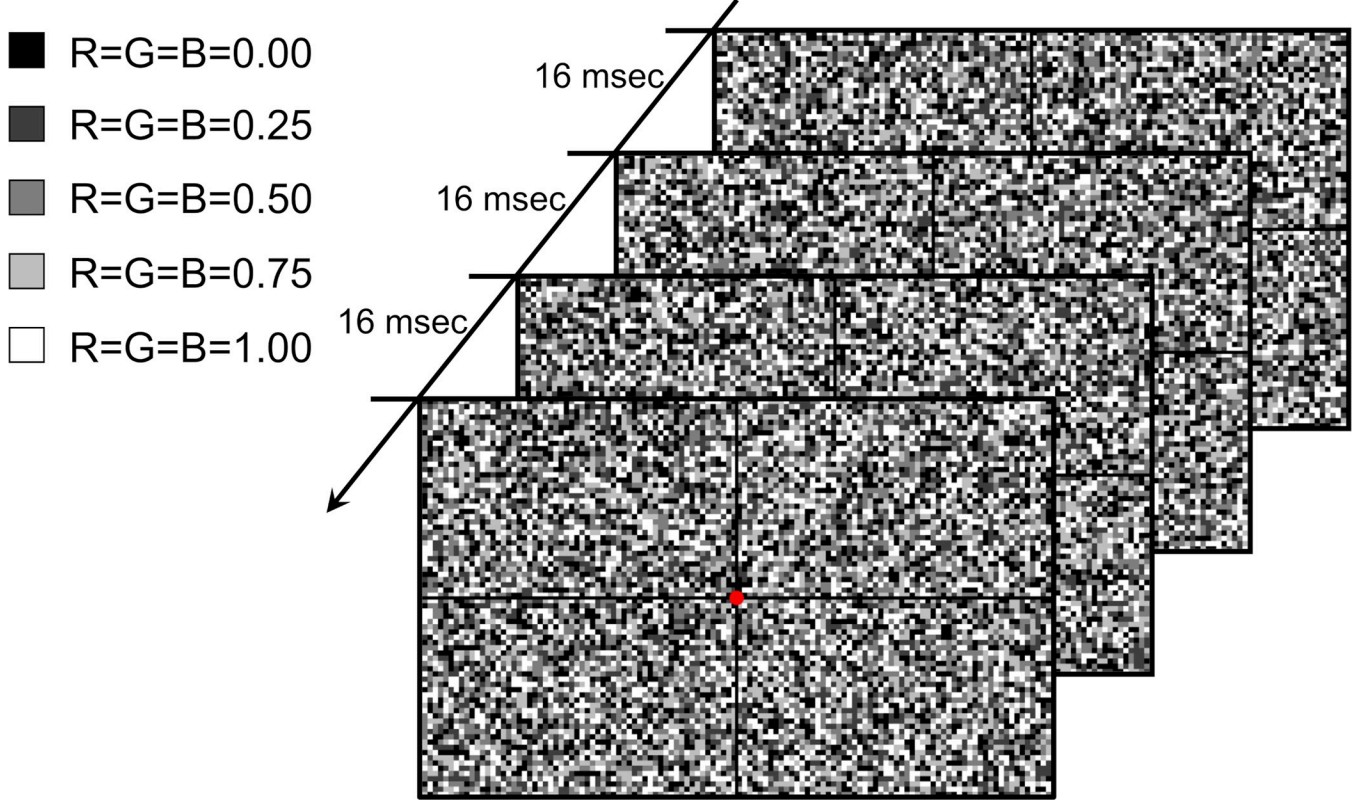

**Fig 2. Procedure for creating CG noise.** (Left side) Five kinds of square dots. R, G, and B indicate a color value between 0 and 1 in the RGB color model. (Right side) Creating a noise-moving image from the noise-still images. CG: computer graphics, R: red, G: green, B: blue.

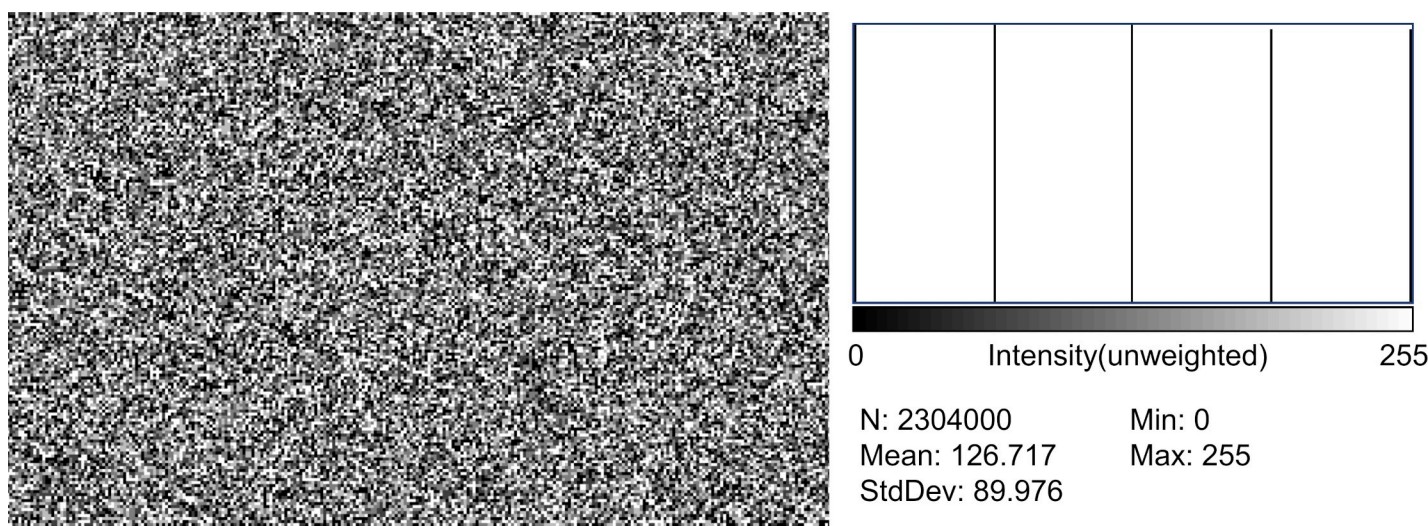

**Fig 3. Still image of CG noise.** (Left side) Still image of CG noise: An image was programmed to have 1920 × 1200 = 2,304,000 pixels to match the pixels of the display screen for examination. (Right side) A histogram of luminance distribution: only five types of luminance values (0, 64, 127, 191, and 255) exist in approximately equal numbers as programmed. N (total pixel count): 2,304,000, Mean (luminance value): 126.717, StdDev (standard deviation): 89.976, CG: computer graphics.

Shenzhen, China), the measured luminance values were 388, 209, 87, 21, and 0.27 cd/m$^2$. Here, the RNG is an algorithm for generating random numbers in a computer program. In this program, the Xorshift method [14] was used.

### Configuration diagram of the examination display screen, PC, and analog TV

The analog and CG noises were displayed on a full screen in the same display, as shown in Fig 4. The noises were switched instantly using a High-Definition Multimedia Interface (HDMI) switcher. Analog TV images were National Television System Committee standard signals, 30 frames/s, with interlaced scanning (60 fields/s). However, when the analog noise was converted to HDMI, the signal reached 60 frames/s, with progressive scanning, as confirmed by a digital camera capable of high-speed shooting.

The examination display was an LCD (CG247X®, EIZO Corp., Hakusan, Japan), measuring 51.84 cm in width (1920 pixels) and 32.4 cm in height (1200 pixels), with a dot pitch of 0.27 mm/pixel and a visual angle of 40.8˚ in width and 28.4˚ in height at a 30-cm viewing distance. The display settings were as follows: maximum brightness, 400 cd/m$^2$; color temperature, native (the native is the monitor's original color); gamma, 2.2; and color gamut, native. A transparent film depicting a red 6-mm diameter fixation point at the center and horizontal and vertical meridians separating the four quadrants was affixed to the display screen.

### Procedure of noise-field test

The examination room had quasi-dark illuminance with 8–9 lux. The viewing distance was set as 30 cm, and all participants were corrected for near vision. For the analog noise, real-time noise was used, instead of noise recordings. The participants were examined one eye at a time,

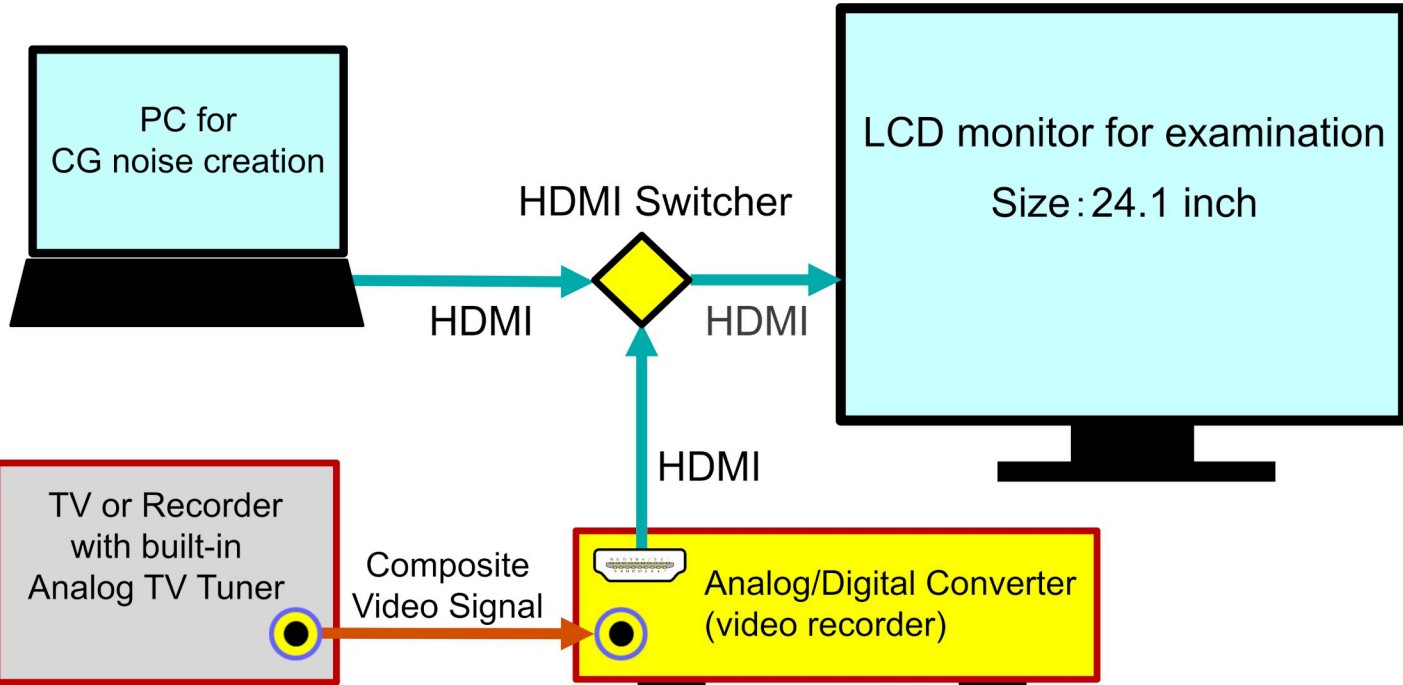

**Fig 4. Diagram of the examination display screen, PC, and analog TV.** The noises were switched instantly using a High-Definition Multimedia Interface (HDMI) switcher. PC: personal computer, CG: computer graphics, LCD: liquid-crystal display.

with the other eye covered with a shielding plate. The analog and CG noises were presented twice in that order. Initially, the participants were asked to fixate on the central point and to report any differences or shadows in the four quadrants. If the perceived shadow differed between the first and second presentations, a third presentation was conducted to confirm reproducibility. If the shadow was inconsistent after four or more presentations, it was deemed "not reproducible." For reproducible shadows, the participants were asked to describe the shadow's range, shape, positional relationship with the fixation point, horizontal and vertical meridians, and any differences between the analog and CG noises on a designated recording chart. The participants who were unaware of or could not reproduce the shadows were categorized as having "no response," and those who were unable to draw shadows were considered maladaptive to the examination.

Additionally, the participants who detected shadows in both the analog and CG noises were asked to identify which noise made the shadows easier to recognize by choosing from the following three options: "analog," "CG," or "same."

The above procedure was performed as one test, and to ensure accuracy, two examiners (i.e., a medical staff member and an ophthalmologist) performed the same test twice. If the results of the first and second tests differed, a third test was performed.

## Participants and procedures

The data of participants who underwent the noise-field test and the visual field test using the Humphrey field analyzer (HFA; Carl Zeiss Meditec Inc., Dublin, CA) at Inoue Eye Clinic between December 2019 and May 2020 were examined retrospectively from May 19, 2020. This study followed the tenets stipulated in the Declaration of Helsinki. Oral informed consent was obtained from all participants in the presence of a nurse as a witness and documented in the medical record. The procedures performed were approved by the Japan Medical Association Ethical Review Committee with an opt-out consent system.

The participants included those with glaucoma, preperimetric glaucoma (PPG), and normal eyes. One eye per case was selected, using the eye with the lower mean deviation (MD) on the HFA if both eyes were eligible. All participants underwent ophthalmic examinations, which included distant corrected visual acuity (DCVA), slit lamp examination, applanation tonometry, dilated funduscopy, and spectral-domain optical coherence tomography (SD-OCT) (RS-3000, Nidek Corp., Gamagori, Japan), and HFA. The exclusion criteria included those with central visual field defects, DCVA of <0.1, chorioretinal disease, retinal artery or vein occlusion, pre-proliferative or proliferative diabetic retinopathy, prior retinal detachment or vitreous surgery, or any history of optic nerve or cranio-orbital diseases causing visual field abnormalities.

The diagnosis of glaucoma was based on the presence of glaucomatous optic disc changes and thinning of macular ganglion cell complex (mGCC) thickness on SD-OCT, which was consistent with visual field abnormalities.

## Criteria for diagnosing glaucomatous visual field abnormalities

The 30–2 Swedish Interactive Threshold Algorithm-standard program on the HFA was performed, excluding unreliable results (false-positive ≥15% or false-negative ≥33%). The glaucomatous visual field abnormality was determined based on the criteria by Anderson–Pattela, which should include one or more of the following conditions [15]: (1) on the pattern deviation probability plot, a cluster of three or more non-edge points with a $P$ value < 5%, including one point or more with a $P$ value < 1% for each superior and inferior hemifield; but, two

points on the nasal edge were included; (2) a pattern standard deviation with a *P* value < 5%; and (3) glaucoma hemifield test results outside normal limits.

The absence of all these criteria indicated a normal visual field, and a normal eye had no abnormal findings in the fundus, no mGCC thinning, and a normal visual field. The normal eye group consisted of the normal eyes of participants, as confirmed during health examinations and those of nonglaucoma patients. Cases with glaucomatous optic disc changes and mGCC thinning but with normal visual fields were classified as PPG cases.

### Thinning of mGCC thickness on SD-OCT

The thinning of the mGCC thickness on SD-OCT was defined as an area measuring <1% of the normative database. A regular normative database was used for the OCT analysis [16, 17]. A long axial length normative database was used for the eyes with an axial length of ≥26 mm or eyes with high myopia of −6.0D or less in phakic eyes. The axial length was measured using IOLMaster (Carl Zeiss Meditec AG, Jena, Germany). The unmeasured axis length was converted to $24 + 0.333 \times |ES|$ mm, with |ES| representing the absolute value of the equivalent spherical value.

### Stage classification of glaucoma

The glaucomatous visual field was classified into the following four stages based on the HFA's MD: M1 (early), MD > −6.0 dB; M2 (moderate), MD ≤ −6.0 dB and MD ≥ −12.0 dB; M3 (advanced), MD < −12.0 dB; and M4 (severe), MD around ≤−20 dB, with no pattern deviation shown [18].

### Agreement in HFA and noise-field test

The abnormal area in the HFA was defined based on the abovementioned clusters of visual field anomalies. In the noise-field test, the abnormal area was determined by the participants' drawings of shadows on paper. In the present study, Mariotte's blind spot was excluded from the abnormal area. The shadows in Mariotte's blind spot appeared as circular or elliptical, straddling the upper and lower hemifields at a position approximately 15˚ temporal to the horizontal meridian, but these shadows were not contiguous with the other shadows.

In terms of the agreement between the HFA and the noise-field test, the procedure reported earlier [4, 7] was used to evaluate whether the patient-drawn shadows were the result of glaucoma. The visual field was divided into the superior and inferior hemifields, and it was determined whether the hemifields with abnormal areas in each test coincided. "Agreement" indicated that both tests showed abnormal areas in the same hemifield. "Partial agreement" showed coincidence in one hemifield and no coincidence in another. "No agreement" indicated that both hemifields with abnormal areas differed entirely. The agreement and partial agreement were defined as the subjective detection of visual field abnormalities using the noise-field test. An example of the "agreement" is shown in Fig 5.

### Image analysis of noises

The analog noise was recorded on a Blu-ray hard disk recorder (BD-HDW80®, Sharp Corp.) and copied to a Blu-ray Disc®. The noise on the Blu-ray Disc was the Moving Picture Experts Group (MPEG)-2 standard video with a resolution of 720 × 480 at 29.97 fps. Using video-editing software (Adobe Premiere Pro version 23.4, Adobe Inc., San Jose, CA) on a PC, 60 sequential still images were extracted from the Blu-ray Disc video without any editing functions, such as brightness adjustment. For the CG noise, 60 still images were produced.

## (A) HFA 30-2, SITA-Standard

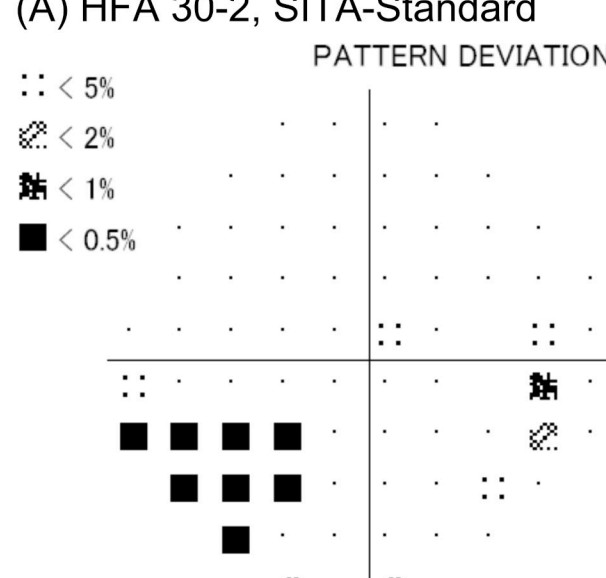

PATTERN DEVIATION

: : < 5%
< 2%
< 1%
■ < 0.5%

MD -2.79 dB, PSD 9.33 dB, VFI 94%

## (B) Recording Chart of Noise-Field Test

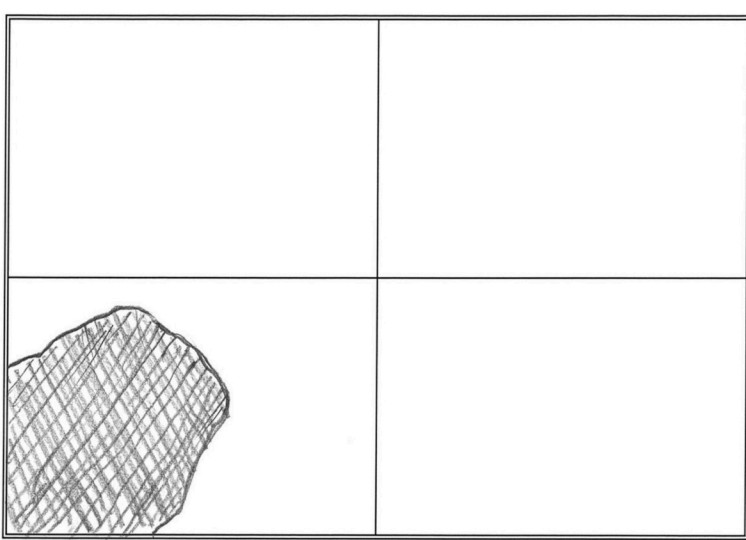

**Fig 5. An example of the "agreement".** The right eye of a 64-year-old man with no subjective symptoms (Stage M1). (A) The pattern deviation probability map in the HFA shows two clusters comprising six points on the inferior nasal region and three points on the inferior temporal region. (B) Recording chart of the noise-field test. The shadow diagram drawn by the patient was judged as showing "agreement" with the HFA. HFA: Humphrey field analyzer, SITA: Swedish Interactive Threshold Algorithm, MD: mean deviation, PSD: pattern standard deviation, VFI: visual field index.

All still images were visually inspected for noise bias. Additionally, the image analysis software ImageJ 1.53 (National Institutes of Health, Bethesda, MD) was used to analyze the still images of the analog and CG noises. Histograms were utilized to calculate the luminance value (gray value: 0–255) of all pixels in a still image, with the luminance value on the X–axis and the number of pixels for each luminance value on the Y-axis. The differences in the luminance distribution and mean luminance value between the analog and CG noises were also examined. The mean luminance values were calculated for each still image. Additionally, the means of the mean luminance values of 60 images were calculated.

### Statistical analysis

For statistical analyses, the McNemar test, chi-square test with adjusted residual analysis, binomial test, one-way analysis of variance, and t-test were performed using SPSS version 29 (IBM Japan, Ltd., Tokyo, Japan). The significance level was set at $P < 0.05$ (two-tailed).

### Results

#### Clinical outcomes

We analyzed 205 eyes with glaucoma, 19 eyes with PPG, and 35 normal eyes. Table 2 shows the participants' demographics.

The agreement between the HFA and the noise-field test is indicated in Table 3, and the detection rates in the noise-field test in the glaucoma group are shown in Table 4. The

**Table 2. Demographic characteristics of the study participants (259 eyes).**

| Characteristics | Value |
|---|---|
| Glaucoma | 205 |
| Sex (male/female) | 94/111 |
| Age, years as the mean ± standard deviation (range) | 71.9 ± 10.1 (35–89) |
| Glaucoma type | |
| POAG | 66 |
| NTG | 119 |
| CACG | 6 |
| SG | 14 |
| Visual field stage classified by mean deviation (MD) | |
| M1: −6.0 dB < MD | 97 |
| M2: −12.0 dB ≤ MD ≤ −6.0 dB | 55 |
| M3: MD < −12.0 dB | 33 |
| M4: Pattern Deviation not shown for severely depressed fields | 20 |
| Preperimetric Glaucoma (PPG) | 19 |
| Sex (male/female) | 8/11 |
| Age, years as the mean ± standard deviation (range) | 63.6 ± 12.1 (37–78) |
| Normal | 35 |
| Sex (male/female) | 7/28 |
| Age, years as the mean ± standard deviation (range) | 57.8 ± 14.2 (28–82) |

POAG: primary open-angle glaucoma, NTG: normal-tension glaucoma

CACG: chronic angle-closure glaucoma, SG: secondary glaucoma

sensitivity values were 80.5% and 65.4% for the CG (95% confidence interval [CI]: 74.4%–85.7%) and analog (95% CI: 58.4%–71.9%) noises, respectively, in the glaucoma group.

As the stage progressed, the detection rate of both the CG and analog noises tended to increase. In stage M3, the detection rates of the CG and analog noises were 93.9% (95% CI: 79.8%–99.3%) and 78.8% (95% CI: 61.1%–91.0%), respectively.

Considering the difference in the detection rate of each noise between the glaucoma stages, the detection rate in stage M1 was significantly lower for the CG noise, whereas the detection

**Table 3. Agreement between the HFA and the noise-field test.**

| | Total eyes | CG noise | | | | analog noise | | | |
|---|---|---|---|---|---|---|---|---|---|
| | | Agree-ment | Partial Agreement | No Agree-ment | No Re-sponse | Agree-ment | Partial Agreement | No Agree-ment | No Re-sponse |
| Normal Visual Field | | | | | | | | | |
| Normal Eyes | 35 | – | – | 1 | 34 | – | – | 0 | 35 |
| PPG | 19 | – | – | 0 | 19 | – | – | 0 | 19 |
| Glaucoma Stage | | | | | | | | | |
| M1 | 97 | 45 | 21 | 0 | 31 | 35 | 14 | 0 | 48 |
| M2 | 55 | 38 | 13 | 0 | 4 | 32 | 12 | 0 | 11 |
| M3 | 33 | 17 | 14 | 0 | 2 | 15 | 11 | 0 | 7 |
| M4 | 20 | 16 | 1 | 0 | 3 | 14 | 1 | 0 | 5 |
| Total | 205 | 116 | 49 | 0 | 40 | 96 | 38 | 0 | 71 |

PPG: preperimetric glaucoma, HFA: Humphrey field analyzer, CG: computer graphics

**Table 4. Detection rates in the noise-field test in the glaucoma group.**

| Glaucoma Stage | Total eyes | Detection by CG noise | | | | Detection by analog noise | | | | McNemar test (CG vs analog) *P* value |
|---|---|---|---|---|---|---|---|---|---|---|
| | | eyes | rate | 95% CI | chi-square test * | eyes | rate | 95% CI | chi-square test * | |
| M1 | 97 | 66 | 68.0% | 57.8%–77.1% | −4.26 (<0.001) a | 49 | 50.5% | 40.2%–60.8% | −4.24 (<0.001) c | <0.001 |
| M2 | 55 | 51 | 92.7% | 82.4%–98.0% | 2.68 (0.007) b | 44 | 80.0% | 67.0%–89.6% | 2.67 (0.008) d | 0.0156 |
| M3 | 33 | 31 | 93.9% | 79.8%–99.3% | 2.13 (0.033) b | 26 | 78.8% | 61.1%–91.0% | 1.77 (0.077) d | 0.0625 |
| M4 | 20 | 17 | 85.0% | 62.1%–96.8% | 0.54 (0.592) a, b | 15 | 75.0% | 50.9%–91.3% | 0.95 (0.341) c, d | 0.5000 |
| M3+M4 | 53 | 48 | 90.6% | 79.3%–96.9% | – | 41 | 77.4% | 63.8%–87.8% | – | 0.0156 |
| Total | 205 | 165 | 80.5% | 74.4%–85.7% | – | 134 | 65.4% | 58.4%–71.9% | – | <0.001 |

The chi-square tests for both CG and analog groups resulted in values of 18.876 (*P* < 0.001) and 18.098 (*P* < 0.001), respectively. CG: computer graphics, CI: confidence interval

*: adjusted residual (*P* value). Each letter denotes a subset of stage categories whose column proportions do not differ significantly from each other at the 0.05 level.

rates in stages M2 and M3 were significantly higher. There were no significant differences in subset a (stages M1 and M4) and subset b (stages M2, M3, and M4). For the analog noise, the detection rate in stage M1 was significantly lower, whereas that in stage M2 was significantly higher. Subset c (stages M1 and M4) and subset d (stages M2, M3, and M4) were not significantly different. The detection rate in stage M4 did not differ significantly from those in other stages but was lower than those in stages M2 and M3 for both CG and analog noises.

In all stages, the CG noise demonstrated superior detection rates as compared to the analog noise, with significant differences confirmed by the McNemar test, except for stages M3 and M4, which had a small number of eyes. When stages M3 and M4 were combined (i.e., in the group with an MD value <−12.0 dB), a significant difference was observed.

The PPG eyes did not exhibit subjective abnormalities with either the CG or analog noise. The specificity values were 97.1% and 100% for the CG and analog noises, respectively, in normal eyes.

There were 134 glaucomatous eyes with subjective abnormalities detected by both the CG and analog noises. Table 5 indicates the ease of shadow recognition in 134 glaucomatous eyes with subjective abnormalities detected by both the CG and analog noises. The CG noise was better in 94 eyes (70.1%), the analog noise was better in 10 eyes (7.5%), and the same answer was obtained in 30 eyes (22.4%). There was a significant difference in the ease of shadow recognition between the CG and analog noises in all stages.

**Table 5. Comparison of the ease of shadow recognition between CG and analog noises.**

| Glaucoma Stage | Total eyes | CG | | same | | analog | | binomial test (CG vs analog) *P* value |
|---|---|---|---|---|---|---|---|---|
| | | eyes | rate | eyes | rate | eyes | rate | |
| M1 | 49 | 37 | 75.5% | 10 | 20.4% | 2 | 4.1% | <0.001 |
| M2 | 44 | 29 | 65.9% | 10 | 22.7% | 5 | 11.4% | <0.001 |
| M3 | 26 | 19 | 73.1% | 5 | 19.2% | 2 | 7.7% | <0.001 |
| M4 | 15 | 9 | 60.0% | 5 | 33.3% | 1 | 6.7% | 0.021 |
| total | 134 | 94 | 70.9% | 30 | 22.4% | 10 | 7.5% | <0.001 |

CG: computer graphics

### Result of image analysis of noises

Sixty still images of the analog noise were inspected, and no bias, such as mottled or linear clumps, was observed in any image. The histograms of the 60 analog noise-still images were analyzed, with 38 showing a normal-distribution-like bell-shaped histogram and 28 with a trapezoid-shaped histogram (Fig 1). All still images encompassed the luminance values ranging from 0 to 255.

As programmed, the CG noise had only five types of luminance values, which are as follows: 0, 64, 127, 191, and 255. The number of pixels for each luminance value in a single still image was measured, and the means for the 60 images were as follows: 460461.8 ± 4066.9 (standard deviation; SD), 461044.3 ± 4292.1 (SD), 460639.1 ± 4692.9 (SD), 461225.1 ± 4138.3 (SD), and 460521.4 ± 4145.3 (SD) in luminance order. A one-way analysis of variance confirmed no statistically significant differences, indicating equal distribution among the five luminance values.

The mean luminance values were 78.009 ± 1.092 (SD) and 127.409 ± 0.412 (SD) for the analog and CG noises, respectively, with the analog noise showing significantly lower luminance values (t-test, $P < 0.001$).

## Discussion

The clinical significance of the noise-field test lies in its ability to convert a patient's negative scotoma into a positive scotoma. As reported previously, an analog noise (TV snow noise) had a higher detection rate than a digital noise (TEC) [12]. However, TEC production was discontinued, and the TV snow noise was less accessible. Therefore, we created a digital noise by CG, compared its properties with those of the analog noise, and found it to have a higher detection rate.

### Homogeneity of noise

In the noise-field test, a noise must exhibit homogeneity, i.e., a uniform distribution and a lack of bias. The test becomes invalid if the noise displays a stripe pattern, stationary dots, or flows in a particular direction, as shown in Fig 6A. Since the analog noise is available in various

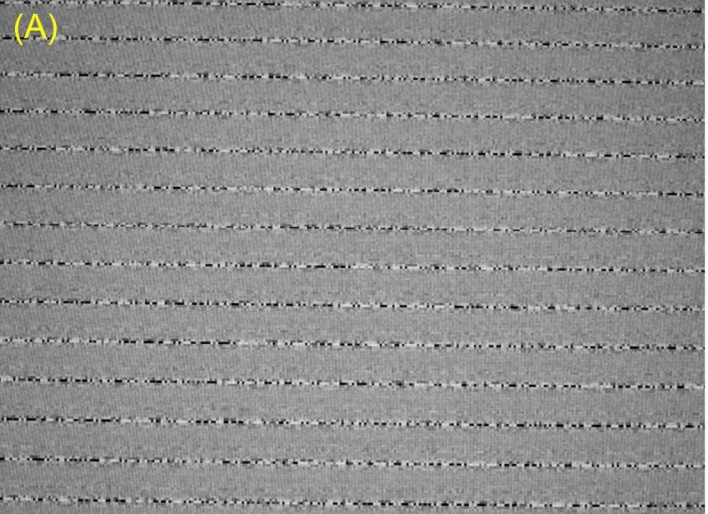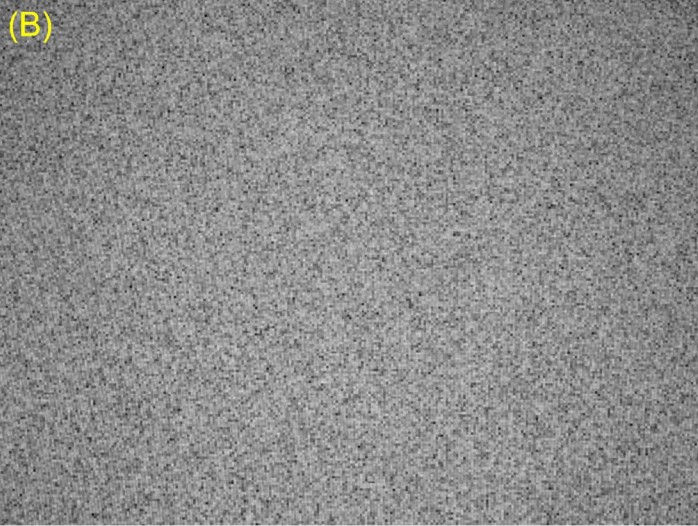

**Fig 6. Difference of noise by random number generation method (display photo).** The noises on the display screen were taken with a digital camera at a fixed shutter speed of 1/5 second. (A) The CG noise generated using LCG shows a stripe pattern. (B) The CG noise generated using the Xorshift method is a homogeneous image. This judgment cannot be made by looking at still images for each frame. CG: computer graphics, LCG: linear congruential generator.

types, we selected the most homogeneous and stable noise through visual inspection. Additionally, given the 100% specificity observed in the analog noise test, the analog noise chosen for this study is considered to exhibit sufficient homogeneity.

For the CG noise, we also visually confirmed the noise generated by the RNG. The computer program utilized the Xorshift method as the RNG. Initially, we generated the CG noise using a linear congruential generator (LCG) [19]. However, when LCG was used under specific conditions, such as dot size, number of colors, and display frequency, inhomogeneous noises, as described earlier, occasionally occurred, and the test was rendered impossible (Fig 6A). However, under the same conditions, when only the RNG was switched to the Xorshift method, a homogeneous noise was obtained (Fig 6B). The CG noise is three-dimensional on the still image (quadratic plane) and time axis (see Fig 2). In the noise produced by LCG, although the still image may appear random, changes on the time axis reveal a lack of uniform distribution, which is considered to be due to a statistical defect in LCGs called the "lattice structure" [20, 21].

Subsequently, several high-quality RNGs were proposed, including the Mersenne twister [22] and Xorshift methods [14]. These RNGs have passed statistical tests for randomness and exhibit no lattice structure. The choice of RNG is crucial in CG noise generation, as the homogeneity of the CG noise varies depending on the RNG. Finally, analyzing still images alone is insufficient to confirm noise homogeneity, and visual inspection of moving images is considered the simplest and most reliable method.

## Difference in properties between analog and CG noises

In the present study, the same monitor was used under the same conditions for both the analog and CG noises. The refresh rates were confirmed to be 60 frames/s for both noises. The CG noise dot size of 2 mm was based on the TEC setting. The size of the analog noise dots could not be determined because the noise consisted of small particles fused and had blurred boundaries.

The analysis of 60 sequential still images revealed differences in noise properties. The analog noise had a low mean luminance value, encompassed all luminance values, and exhibited unclear differences between high and low luminescent spots. Contrarily, the CG noise assigned each luminescent spot a luminance value divided into five equal parts of the maximum luminance value, resulting in a clear distinction between high and low luminance spots, which allows the identification of the difference in flicker between the abnormal and normal areas of the visual field in the CG noise, thereby contributing to the difference in subjective detection rates.

## Differences in detection rates between glaucoma stages

The detection rates of the CG and analog noises tended to increase as the glaucoma stage progressed; however, no significant difference was observed above stage M2. The detection rate in stage M4 did not differ significantly from that of the other stages; however, it was lower than the detection rates in stages M2 and M3. The patients in stage M4 could hardly see anything except the center, and as the normal area decreases, it may be difficult to detect the presence or absence of noise flicker. Consequently, the detection rate might have been reduced.

## Advantages of our noise-field test developed in this study

We have developed the application software that generates noises by CG, which is versatile and hardware-independent, maximizing the benefits of the noise-field test. This software

works on Windows® operating system, and future porting to macOS® (Apple Inc., Cupertino, CA) is feasible.

The advantage of our CG noises is that the dot size, color distribution, and frequency of the noise can be changed arbitrarily and instantaneously. The frequency depends on the display's performance and can be up to 1000 fps in the computer program. We have confirmed that a 360-fps monitor can operate at its intended capacity. However, in this study, the CG noise was set to 2 mm for dot size, five monochromatic tones for color, and 60 fps for frequency based on the TEC setting.

One of our innovations is to draw the fixation point and meridians that divide the four quadrants by CG. However, in the present study, a transparent film with the fixation point and meridians was affixed to the display to create the same condition as that used for the analog noise. Including the fixation point and meridians enhances the participant's ability to focus on the fixation point efficiently and recognize the positional relationship between shadows and meridians.

Ideally, a larger display is preferred for the noise-field test to assess the entire visual field at once. A display with a width of $\geq$34.6 cm is necessary for the participants to obtain a central 30-degree visual angle at a viewing distance of 30 cm. The display used in this study provided a visual angle of 40.8˚ in width and 28.4˚ in height, allowing for a wide inspection area.

## Comparison with previous reports

Adachi et al. compared the characteristics of analog (TV snow noise) and digital (TEC) noises in 136 eyes with glaucoma. The detection rate of the analog noise (85.3%) was higher than that of TEC (fine type, 75.0%; coarse type, 71.3%). In the Aulhorn classification–Greve modified stage III or later, the detection rate of the analog noise was 100% [12]. In our results, the detection rate of the analog noise was 65.4%, and even in stage M2, the detection rate was 80.0%, which was the highest. These results were relatively lower than the previous study's results. The comparisons with Adachi's report cannot be made simply because of the differences in the assessment methods, glaucoma staging, and other factors. One of the different conditions was that their reports used cathode ray tube (CRT) displays for both noises. The most significant difference between LCDs and CRTs is that the LCD monitors present images continuously (hold type), whereas CRTs present images in a flash style (impulse type). Due to this structural difference, LCDs have slow response times of luminance changes on one pixel [23, 24]. For this reason, in pattern-reversal visually evoked potentials using displays, the latency is reported to be longer for LCDs than for CRT displays [25, 26]. For the noise-field tests, a CRT display may be more suitable. However, CRT displays are currently not available.

Recently, iPad-based noise-field perimeters were reported to show a detection rate of 91.2% in glaucomatous eyes (52/97) [27]. However, the distribution of glaucoma stages differs from our study. Although the relationship between the glaucoma stage and detection rate has not been clearly described, the detection rate in the early stage with an MD value of $\geq$−5.0 dB was 69.2% (9/13). The detection rate of our CG noise in the early stage (>−6.0 dB) was 68.0% (66/97), which was almost equivalent with that of the previous study. Although tablet PCs, such as iPads, are portable and convenient to use anywhere, their small size is a disadvantage. The area examined using this device was divided into four quadrants, and the fixation point was changed to one of the four corners of the image in each quadrant. The entire visual field could not be viewed at once, and it may be difficult for the participants to observe the differences among the four quadrants. The participants may be aware of Mariotte's blind spot in the noise-field test [28], but evaluating the difference between Mariotte's blind spot and scotomas

in the vicinity is sometimes difficult. Therefore, a larger display is still considered more appropriate for the noise-field test.

## Limitations and future work

This study has some limitations. First, it has a single-center design, and the patient population was relatively small. Prospective multicenter studies with a large number of participants are needed.

With regard to the problems of the examination procedure, to keep the process constant, the tests were performed in the order of analog and CG noise. Familiarity with the test might have increased the detection rate of the noise presented later. However, the examinees watched a noise at least three times in one test and six times in two tests. Thus, it seems unlikely that the detection rate of the noise presented second would be higher. However, this issue cannot be denied and should be confirmed in the future.

With regard to the method of determining agreement in the HFA and the noise-field test, a comparison of each upper and lower hemifield was used to determine whether the patient-drawn shadows were due to glaucoma in this study. The reason is that glaucomatous visual field defects often cross the vertical meridian because the defects occur along the retinal nerve fiber layer. The same concept has been used to determine the consistency of mGCC thinning on the OCT and visual field abnormalities in patients with glaucoma. When quantifying patient-perceived shadows or checking for other diseases, the quadrant or more detailed zonal evaluation will be useful.

Noise-field tests rely on subjective perception, and the reliability of the shadows drawn by the participants can only be evaluated by comparing them with other visual field tests. It is important to note that this test is not superior to standard automated perimetry.

The primary purpose of our noise-field test is to make patients with no subjective symptoms to be aware of visual field abnormalities, increase their awareness of their clinical manifestations, and motivate them to continue their treatment. Our test showed that the detection rate was about 80% for all patients with glaucoma and 90% for the moderate and later stages, so it is presumed that this purpose has been achieved to some extent. Our test also has high specificity and may serve as a screening and self-check test for glaucoma. However, the detection rate was approximately 70% in the early glaucomatous stage, and it did not reach 100% even in the advanced stages. To enhance the detection rate of the noise-field test, further studies are needed to optimize the parameters of CG noise, including dot size, color distribution, and frequency.

## Conclusions

We created a homogeneous random noise-moving image using CG. The CG noise is more effective than the analog noise in evaluating the subjective perception of visual field abnormalities in patients with glaucoma. It may be useful in improving patients' understanding of their clinical signs and symptoms.

## Supporting information

**S1 Video. Our CG noise is directly displayed on the monitor by the computer program.** This video is a conversion of the noise to MPEG4 format, and the quality of the image is relatively poor. CG: computer graphics.
(MP4)

## Author Contributions

**Conceptualization:** Arata Inoue, Naoyuki Maeda, Chota Matsumoto.

**Data curation:** Arata Inoue, Eiko Koike.

**Formal analysis:** Arata Inoue, Naoyuki Maeda, Chota Matsumoto.

**Funding acquisition:** Arata Inoue.

**Investigation:** Arata Inoue, Eiko Koike.

**Methodology:** Arata Inoue, Naoyuki Maeda, Chota Matsumoto.

**Project administration:** Arata Inoue, Chota Matsumoto.

**Resources:** Arata Inoue.

**Software:** Arata Inoue.

**Supervision:** Chota Matsumoto.

**Validation:** Arata Inoue.

**Visualization:** Arata Inoue.

**Writing – original draft:** Arata Inoue.

**Writing – review & editing:** Arata Inoue, Eiko Koike, Naoyuki Maeda, Chota Matsumoto.

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
