## [Decision Letter · Decision Letter 0]

8 Dec 2023

PONE-D-23-35504Subjective perception of visual field defects using random noise-moving images in patients with glaucoma: A comparison of computer graphics and analog noisesPLOS ONE

Dear Dr. Inoue,

Thank you for submitting your manuscript to PLOS ONE. After careful consideration, we feel that it has merit but does not fully meet PLOS ONE’s publication criteria as it currently stands. Therefore, we invite you to submit a revised version of the manuscript that addresses the points raised during the review process.

We look forward to receiving your revised manuscript.

Kind regards,

Ryo Asaoka

Academic Editor

PLOS ONE

2. In the ethics statement in the Methods, you have specified that verbal consent was obtained. Please provide additional details regarding how this consent was documented and witnessed, and state whether this was approved by the IRB.

3. We note that you have a patent relating to material pertinent to this article. Please provide an amended statement of Competing Interests to declare this patent (with details including name and number), along with any other relevant declarations relating to employment, consultancy, patents, products in development or modified products etc. Please confirm that this does not alter your adherence to all PLOS ONE policies on sharing data and materials, as detailed online in our guide for authors http://journals.plos.org/plosone/s/competing-interests by including the following statement: "This does not alter our adherence to  PLOS ONE policies on sharing data and materials.” If there are restrictions on sharing of data and/or materials, please state these. Please note that we cannot proceed with consideration of your article until this information has been declared.

Additional Editor Comments:

Please respond to the points raised by the reviewers carefully.

Reviewers' comments:

Reviewer's Responses to Questions

**Comments to the Author**

1. Is the manuscript technically sound, and do the data support the conclusions?

Reviewer #1: Yes

Reviewer #2: Yes

Reviewer #3: Partly

2. Has the statistical analysis been performed appropriately and rigorously? 

Reviewer #1: Yes

Reviewer #2: Yes

Reviewer #3: Yes

3. Have the authors made all data underlying the findings in their manuscript fully available?

Reviewer #1: Yes

Reviewer #2: Yes

Reviewer #3: No

4. Is the manuscript presented in an intelligible fashion and written in standard English?

Reviewer #1: No

Reviewer #2: Yes

Reviewer #3: Yes

5. Review Comments to the Author

Reviewer #1: This paper provides a detailed study of the usefulness of CG noise in glaucoma diagnosis. The authors would like to consider the following revisions.

1. There is no mention of recruiting healthy volunteers. Which eye did the authors use as the sample, left or right?

2. In the Table 4, how about using contingency table analysis to compare detection rates at each glaucoma stage?

3. The authors should discuss the fact that the detection rate of M4 is lower than that of M3.

Reviewer #2: The authors have developed a novel approach using the random noise-moving images which enables glaucoma patients to aware of the presence of visual field abnormalities. In this paper, they introduced a customized noise images using computer graphics (CG) instead of conventional analog noise. Furthermore, the authors showed that their approach is more effective than the analog noise in evaluating the subjective perception of visual field abnormalities in glaucoma patients.

In general, the methodology is reasonable, and there are no apparent issues with the conclusions drawn from the results. However, further explanation should be added in the discussion regarding why the CG images yielded better results compared to the analog noise. While the authors attributed the improvement to the higher luminance contrast between frames, it would be worthwhile to consider other factors, such as the followings.

The authors note that the output video utilizing CG images maintains a frame rate of 60 frames per second. However, there is no mention of the refresh rate for the analog noise, which was transmitted through HDMI via the A/D converter. It is important to clarify this point, as even though the HDMI transmission standard may support 60 frames per second, the actual displayed video may depend on the underlying NTSC signal, which is approximately 30 frames per second. If the refresh rate differs, albeit slightly, there is a potential for the effects of temporal summation.

The rationale behind using 2mm square dots in CG image is not explained. Considering that the analog noise consists of smaller dots, the difference in the size potentially affects the spatial summation.

Minor point:

With regard to the HFA SITA test, a false-positive rate of less than 15% is considered as reliable result.

Reviewer #3: The authors provide a comparison between static analog noise and a digital approximation of the same noise in terms of making glaucoma patients aware of their glaucomatous defect. This investigation is based on a previously described phenomenon, whereby patients with glaucoma report changes in the way they perceive the analog noise in areas of the visual field affected by the disease.

The manuscript is generally well written, although the English language should be revised. Some sentences seem disconnected or not phrased the way the authors probably intended (i.e. the "Purpose" section in the abstract, "Therefore.." is probably non the best connector; a lot of unproven speculations are treated with certainty, such as Lines 47-50 and 428-429, where "will" should really be "may".)

In general, I struggle to see the benefit of fixating on analog noise for this specific detection task. A lot more work has been done investigating how noise (digital or otherwise) affects perception in glaucoma (https://tvst.arvojournals.org/article.aspx?articleid=2778598, https://iovs.arvojournals.org/article.aspx?articleid=2770281), which could have been used as the starting point. For example, it would have been interesting to show how modifying certain parameters of the noise affected the perception (or identification) of the defect. For example, one leading theory is that the high frequency cortical filter mechanisms are the first ones to lose sensitivity in glaucoma (see Part II in https://jov.arvojournals.org/article.aspx?articleid=2121924 and https://iovs.arvojournals.org/article.aspx?articleid=2163035). This alternative framing would have also given the authors the chance to develop an actual test where participants could have modified some parameters of the noise until the defect became apparent (i.e. a simple detection task, like in perimetry). This would have clarified a lot of the "non reproducible" cases.

One methodological issue is the fact that the authors do not seem to have randomized the presentation of their noise stimuli. The analog signal was always presented first. This means that the increased detection rate could be the result of bias, i.e. patients able to see their defect with the analog noise were "primed" to identify the same defect with the digital noise.

The agreement is also calculated in a perplexing way, since it could have at least considered quadrants rather than hemifields. Moreover, the way the patients were asked to report the extent of their defect seems very imprecise. Alternative methods could have been used, such as a touchpad, where patients would have had the chance to outline the defects as they were seeing it during the test.

Additional care should be taken when considering patients with advanced disease, because a defect will be effectively present everywhere in their field. Since this test only asks patients to compare their perception of relative defects in contrast to the rest of their VF, it might actually lose sensitivity in these patients.

Finally, the scope of the investigation is unfocused. The authors initially state their goal is to make patients aware of their defect but then the manuscript gradually shifts into an assessment of a potential diagnostic test. Neither of these scopes seems to be extensively and sufficiently explored.

6. PLOS authors have the option to publish the peer review history of their article (what does this mean?). If published, this will include your full peer review and any attached files.

Reviewer #1: No

Reviewer #2: No

Reviewer #3: No

---

## [Author Response · Author response to Decision Letter 0]

7 Feb 2024

Reviewer #1:

Comment #1-1:

1. There is no mention of recruiting healthy volunteers. Which eye did the authors use as the sample, left or right?

Author's Response:

Thank you for the useful comment. The normal eye group consisted of the normal eyes of participants as confirmed during health examinations and those of nonglaucoma patients (line 206). We used the lower MD value to select the left and right eyes. The word" prioritizing" has been changed to" using" in the revised manuscript (line 185).

 

Comment #1-2:

2. In the Table 4, how about using contingency table analysis to compare detection rates at each glaucoma stage?

Author's Response:

Thank you for your valuable feedback. Based on the suggestion of reviewer 2, we have changed the exclusion criteria. Accordingly, we have excluded 12 eyes with glaucoma from the study. We conducted a new analysis.

Furthermore, as per the suggestion, a new group of stage M3+M4 was added, and a contingency table of six groups was created. In addition, we analyzed the difference in detection rate between stages using the chi-square test with adjusted residual analyses for each CG noise and analog noise, as shown in the cross tables.

Because the contingency table and the cross tables would be hard to understand if presented separately, we have summarized this information in the revised Table 4.

Comment #1-3:

3. The authors should discuss the fact that the detection rate of M4 is lower than that of M3.

Author's Response:

Thank you for your insightful comment. Per your suggestion, we have analyzed the differences in detection rates between glaucoma stages and added the relevant findings to the Results and Discussion sections of the revised manuscript (lines 295, 386).

The patients in stage M4 could hardly see anything except the center, and as the normal area decreases, it may be difficult to detect the presence or absence of the noise flicker. Consequently, the detection rate might have been reduced, although there were no statistically significant differences.

Reviewer #2:

Comment #2-1:

The authors note that the output video utilizing CG images maintains a frame rate of 60 frames per second. However, there is no mention of the refresh rate for the analog noise, which was transmitted through HDMI via the A/D converter. It is important to clarify this point, as even though the HDMI transmission standard may support 60 frames per second, the actual displayed video may depend on the underlying NTSC signal, which is approximately 30 frames per second. If the refresh rate differs, albeit slightly, there is a potential for the effects of temporal summation.

Author's Response:

Thank you for the insightful comments. The NTSC signal is 30 frames/s with interlaced scanning. More precisely, to draw one frame, the electron gun scans the horizontal lines twice (once for odd-numbered lines and once for even-numbered lines). Each scan is called a "field," with a speed of 60 fields/s. One frame consists of two fields, and 60 fields/s equals 30 frames/s.

However, when the NTSC noise is converted to HDMI, it becomes 60 frames/s with progressive scanning. We used a digital camera capable of high-speed shooting (960 images/s) to capture the noise-moving image directly on the display and examined the changes in the sequential still images. We confirmed that the sequential still images changed every 16 images at 960 images/s and every 4 images at 240 images/s. These results indicate that the noise-moving image is 60 frames/s.

In regular NTSC videos, there is a set of odd and even fields, and the bone core of the still images in these two fields remains unchanged. However, the noise is random, and there is no relationship between the odd and even fields; therefore, it is presumed that a field equals a frame.

Therefore, although it was easy to set the CG noise to 30 frames/s, we intentionally set it to 60 frames/s. We have added this information to the revised manuscript (lines 139, 376).

Comment #2-2:

The rationale behind using 2mm square dots in CG image is not explained. Considering that the analog noise consists of smaller dots, the difference in the size potentially affects the spatial summation.

Author's Response:

Thank you for your useful comment. The CG noise dot size of 2 -mm was determined based on the Tübingen electronic campimeter (TEC) setting (line 401). The size of the analog noise dots could not be determined because the noise consisted of small particles fused and had blurred boundaries. We have revised the text and added this information to the Methods and Discussion sections of our revised manuscript (lines 114, 376). 

To enhance the detection rate of the noise-field test, further studies are needed to optimize the parameters of CG noise, including dot size, color distribution, and frequency.

Comment #2-3:

Minor point:

With regard to the HFA SITA test, a false-positive rate of less than 15% is considered as reliable result.

Author's Response:

Thank you for your useful remark. As per your suggestion, we have changed the exclusion criteria to "false-positive ≥15% or false-negative ≥33%" in the revised manuscript (line 198).

Consequently, we have excluded 14 eyes (12 with glaucoma, 1 with normal-tension glaucoma, and 1 with normal eyes) from the study. We performed all analyses again; however, these modifications did not affect the conclusions of the study.

Reviewer #3:

Comment #3-1:

The manuscript is generally well written, although the English language should be revised. Some sentences seem disconnected or not phrased the way the authors probably intended (i.e. the "Purpose" section in the abstract, "Therefore.." is probably non the best connector; a lot of unproven speculations are treated with certainty, such as Lines 47-50 and 428-429, where "will" should really be "may".)

Author's Response:

Thank you for your pertinent remarks. According to your suggestion, we have made the necessary revisions to the manuscript (lines 23, 47–48, 476).

Comment #3-2:

In general, I struggle to see the benefit of fixating on analog noise for this specific detection task. A lot more work has been done investigating how noise (digital or otherwise) affects perception in glaucoma (https://tvst.arvojournals.org/article.aspx?articleid=2778598, https://iovs.arvojournals.org/article.aspx?articleid=2770281), which could have been used as the starting point. For example, it would have been interesting to show how modifying certain parameters of the noise affected the perception (or identification) of the defect. For example, one leading theory is that the high frequency cortical filter mechanisms are the first ones to lose sensitivity in glaucoma (see Part II in https://jov.arvojournals.org/article.aspx?articleid=2121924 and https://iovs.arvojournals.org/article.aspx?articleid=2163035). This alternative framing would have also given the authors the chance to develop an actual test where participants could have modified some parameters of the noise until the defect became apparent (i.e. a simple detection task, like in perimetry). This would have clarified a lot of the "non reproducible" cases.

Author's Response:

Thank you for your insightful comment. The noise-field test began with the analog noise of TV by Professor Aulhorn in 1987, and the usefulness of analog noise has been reported in many previous cases. We investigated the analog noise to clarify how it differed from the CG noise created in this study.

Thank you for all of the useful suggestions regarding the parameters of CG noise. We will use the four papers you presented in our future research and evaluate these parameters.

1. Srinivasan R, Turpin A, McKendrick AM. Developing a screening tool for areas of abnormal central vision using visual stimuli with natural scene statistics. Transl Vis Sci Technol. 2022;11(2): 34. https://tvst.arvojournals.org/article.aspx?articleid=2778598

2. Liu R, Kwon M. Increased equivalent input noise in glaucomatous central vision: Is it due to undersampling of retinal ganglion cells? Invest Ophthalmol Vis Sci. 2020 61(8): 10. https://iovs.arvojournals.org/article.aspx?articleid=2770281

3. Pan F, Swanson WH. A cortical pooling model of spatial summation for perimetric stimuli. J Vis. 2006;6(11): 1159-71. https://jov.arvojournals.org/article.aspx?articleid=2121924

4. Swanson WH, Felius J, Pan F. Perimetric defects and ganglion cell damage: Interpreting linear relations using a two-stage neural model. Invest Ophthalmol Vis Sci. 2004;45(2): 466-72. https://iovs.arvojournals.org/article.aspx?articleid=2163035

Comment #3-3:

One methodological issue is the fact that the authors do not seem to have randomized the presentation of their noise stimuli. The analog signal was always presented first. This means that the increased detection rate could be the result of bias, i.e. patients able to see their defect with the analog noise were "primed" to identify the same defect with the digital noise.

Author's Response:

Thank you for your insightful remarks. To keep the process constant, we performed the tests in the order of analog and CG noise. As you indicated, familiarity with the test might lead to a higher detection rate of the noise presented later.

We agree with the reviewer that the description in the manuscript was insufficient. Two examiners (a medical staff member and an ophthalmologist) performed the same test twice to ensure accuracy. We have added this information to the revised manuscript (line 173).

The examinees were presented twice for analog and CG noise in one test and asked which was easier to understand compared with analog and CG noise. The examinees watched the noise at least three times in one test and six times in two tests. Consequently, it seems unlikely that the detection rate of the noise presented second would be higher. However, because this issue cannot be denied, we have included it as a limitation of the study (line 446).

Comment #3-4:

The agreement is also calculated in a perplexing way, since it could have at least considered quadrants rather than hemifields. Moreover, the way the patients were asked to report the extent of their defect seems very imprecise. Alternative methods could have been used, such as a touchpad, where patients would have had the chance to outline the defects as they were seeing it during the test.

Author's Response:

Thank you for the insightful comment. We agree that our explanation was insufficient; thus, we will explain our concept here.

The purpose of the agreement in this study was to determine whether the patient-drawn shadows were due to glaucoma and was not quantitative. Glaucomatous visual field defects often cross the vertical meridian because the defects occur along the retinal nerve fiber layer. Therefore, a judgment was made by comparing each of the upper and lower hemifields, a technique that was also used in previous studies [4],[7]. The same concept has been used to determine the consistency of mGCC thinning on the OCT and visual field abnormalities in glaucoma patients.

There were some errors in the patient-drawn shadows, and the visual field abnormalities in the HFA were represented by clusters on the pattern deviation probability plot, which were just one indication of glaucomatous visual field impairment and contained some measurement errors. The common feature of the shadows and the clusters is thought to be that they exist in the same hemifield.

Patient-drawn shadows were recorded in each quadrant; however, we considered reproducibility and whether they crossed the horizontal meridian to be important. As you pointed out, the quadrant or more detailed zonal evaluation will be useful when quantifying patient-perceived shadows or checking for other diseases. In the revised manuscript, we have added this information to the Methods section and the list of limitations (lines 230, 452).

A touchscreen response is also a good idea. As compared with a simple response of visibility or invisibility in the general visual field test, the task of describing the shadows outside the fixation point on the touchscreen while maintaining fixation and viewing distance on the screen is presumed to be challenging for the examinees. In this study, we used the method in which the examinees described the shadow on paper. After developing appropriate software for touchscreen use, we want to determine which method is better.

Comment #3-5:

Additional care should be taken when considering patients with advanced disease, because a defect will be effectively present everywhere in their field. Since this test only asks patients to compare their perception of relative defects in contrast to the rest of their VF, it might actually lose sensitivity in these patients.

Author's Response:

Thank you for this pertinent comment. The differences in the detection rates between glaucoma stages were analyzed and added to the Results and Discussion sections of the revised manuscript (lines 295, 386).

As you mentioned, patients in stage M4 could hardly see anything except the center, and as the normal area decreases, it may be difficult to detect the presence or absence of the noise flicker. Consequently, the detection rate might have been reduced, although no statistically significant differences were detected.

Comment #3-6:

Finally, the scope of the investigation is unfocused. The authors initially state their goal is to make patients aware of their defect but then the manuscript gradually shifts into an assessment of a potential diagnostic test. Neither of these scopes seems to be extensively and sufficiently explored.

Author's Response:

Thank you for pointing this out. We agree that our explanation was insufficient; therefore, we have made the necessary revisions to the manuscript.

We believe that there are two main ways of making an early diagnosis of glaucoma.

The first way is to diagnose early structural changes using techniques such as OCT, and the second is to diagnose early functional changes using precise visual field tests and new algorithms. All procedures are performed professionally in a medical facility. In other words, it is a prerequisite for the patient to visit a medical facility.

Another method of early diagnosis is that patients who are usually asymptomatic visit a medical facility and are diagnosed with glaucoma. Therefore, it is necessary to encourage patients to visit medical facilities. One way of doing this is to make patients aware of their visual field abnormalities by performing the noise-field test.

The noise-field test is a screening test that does not require the accuracy of OCT or precision visual field testing; however, it should have high sensitivity and specificity. In this sense, our study examines the diagnostic power of the noise-field test. To enhance the detection rate of the noise-field test, further studies are needed to optimize the parameters of CG noise, including dot size, color distribution, and frequency.

We have made the necessary text corrections in the Introduction and Discussion sections of the revised manuscript (lines 47, 463-471).

Again, we thank you for your careful review of our manuscript and the thoughtful comments provided.

---

## [Decision Letter · Decision Letter 1]

2 May 2024

Subjective perception of visual field defects using random noise-moving images in patients with glaucoma: A comparison of computer graphics and analog noises

PONE-D-23-35504R1

Dear Dr. Arata Inoue,

We’re pleased to inform you that your manuscript has been judged scientifically suitable for publication and will be formally accepted for publication once it meets all outstanding technical requirements.

Kind regards,

Fidan Aghayeva

Academic Editor

PLOS ONE

Additional Editor Comments (optional):

Dear Dr. Arata Inoue,

The presented study is an original study, the most previously raised concerns and comments have been addressed.  According to one of the reviewers, there are still several unresolved issues, such as the uncertainty regarding the usefulness of the test and the lack of randomisation in the testing sequence. However, revision with changes and acknowledging the limitations improved the manuscript and it could be accepted as a research article with an assessment of a potential diagnostic test. Furthermore, there is a patent for a method and device for diagnosing visual field abnormalities (patent No. JP7339900) and an amended statement of Competing Interests to declare this patent (with details including name and number) was provided.

Whilst reviewer 3 has provided additional comments alongside their decision, no revisions for your manuscript are required, as the main raised concerns and comments have been adressed.

Reviewers' comments:

Reviewer's Responses to Questions

**Comments to the Author**

1. If the authors have adequately addressed your comments raised in a previous round of review and you feel that this manuscript is now acceptable for publication, you may indicate that here to bypass the “Comments to the Author” section, enter your conflict of interest statement in the “Confidential to Editor” section, and submit your "Accept" recommendation.

Reviewer #1: All comments have been addressed

Reviewer #2: All comments have been addressed

Reviewer #3: (No Response)

2. Is the manuscript technically sound, and do the data support the conclusions?

Reviewer #1: Yes

Reviewer #2: Yes

Reviewer #3: Partly

3. Has the statistical analysis been performed appropriately and rigorously? 

Reviewer #1: Yes

Reviewer #2: Yes

Reviewer #3: Yes

4. Have the authors made all data underlying the findings in their manuscript fully available?

Reviewer #1: Yes

Reviewer #2: Yes

Reviewer #3: No

5. Is the manuscript presented in an intelligible fashion and written in standard English?

Reviewer #1: Yes

Reviewer #2: Yes

Reviewer #3: Yes

6. Review Comments to the Author

Reviewer #1: (No Response)

Reviewer #2: (No Response)

Reviewer #3: The authors did not fully address my comments, but I understand some of them have to do with limitations in the study design and cannot be addressed with nothing more than an acknowledgement.

Most of the issues still stand, mainly the uncertainty regarding the usefulness of the test and the lack of randomisation in the testing sequence.

The authors mention that they wanted to focus on hemifield and imply that analyzing quadrants would have focused the attention on diseases other than glaucoma. I do not uderstand how this is possible, given the current targeting of sample of patients. My suggestion was to increase localization of the defect. For example, a nasal step from glaucoma would result in only one quadrant being affected.

The changes to the conclusion have however improved the focus of the manuscript and highlight the potential as a self-screening test, while acknowledging the limitations.

7. PLOS authors have the option to publish the peer review history of their article (what does this mean?). If published, this will include your full peer review and any attached files.

Reviewer #1: No

Reviewer #2: No

Reviewer #3: No

---

## [Editor Report · Acceptance letter]

7 May 2024

PONE-D-23-35504R1 

PLOS ONE

Dear Dr. Inoue, 

I'm pleased to inform you that your manuscript has been deemed suitable for publication in PLOS ONE. Congratulations! Your manuscript is now being handed over to our production team.

Kind regards, 

on behalf of

Dr. Fidan Aghayeva 

Academic Editor

PLOS ONE